# Chance-Constrained Inference for Hallucination Risk Control in Large Language Models

## Abstract

Large language models generate outputs stochastically and may produce fluent but invalid responses, including factual hallucinations. Existing mitigation strategies reduce average error rates but do not provide explicit control over the *frequency* of such failures under repeated use. We formulate inference as a deployment-time risk control problem and introduce *chance-constrained inference* (CCI), which directly bounds the probability of hallucinations among accepted generations. Hallucinations are modeled as stochastic constraint violations, and we show that confidence-based selective prediction does not, in general, imply probabilistic risk guarantees. Our contribution is methodological rather than statistical: we recast inference-time generation as a sequential feasibility certification problem under stochastic decoding, using known anytime-valid concentration bounds to provide deployment-time guarantees with adaptive sampling and principled abstention. Experiments on controlled prompts and on standard QA benchmarks (TriviaQA, HotpotQA, TruthfulQA) with a semantic LLM-as-judge verifier demonstrate reliable three-way certification (Feasible / Infeasible / Undecided), stability under verifier noise, and safer composition under repeated use, while confidence-based and self-consistency baselines fail to provide consistent guarantees.

## 1 Introduction

Large language models (LLMs) are increasingly deployed in applications such as question answering, decision support, and multi-step reasoning. Despite strong average performance, LLMs remain unreliable in a critical respect: they may generate fluent but invalid outputs, including factual hallucinations and constraint violations (Maynez et al., 2020; Zhang et al., 2023). In many applications, even infrequent failures are unacceptable, particularly when models are queried repeatedly or embedded in agentic systems.

Most existing mitigation strategies aim to reduce average error rates. Retrieval-augmented generation (Lewis et al., 2020), alignment via human feedback, and post-hoc hallucination detection (Farquhar et al., 2024) improve empirical accuracy but do not provide explicit control over how often failures occur under stochastic decoding. A closely related line of work is selective prediction and abstention (El-Yaniv & Wiener, 2010; Geifman & El-Yaniv, 2017), which rejects low-confidence outputs based on uncertainty estimates. While effective in supervised classification, confidence-based rejection does not generally provide probabilistic guarantees for generative models, where decoding randomness induces variability even for fixed inputs.

A key limitation of prior approaches is that hallucination is implicitly treated as a deterministic failure to be eliminated. In practice, modern decoding procedures such as temperature or nucleus sampling make generation inherently stochastic (Holtzman et al., 2020), rendering hallucination a random event whose probability depends on the input, model, and decoding strategy. As a result, reliability must be defined and enforced at the level of the *generation distribution*, rather than on individual outputs.

In contrast, decision making under uncertainty in fields such as operations research and control is commonly framed using *chance constraints*, which require constraints to hold with high probability rather than almost surely (Prékopa, 1995; Calafiore & Campi, 2006). Chance constraints provide interpretable and adjustable guarantees on failure frequency while avoiding overly conservative behavior.

In this work, we bring chance-constrained optimization to large language model inference. We formulate hallucination control as a deployment-time risk management problem and introduce *chance-constrained inference* (CCI), which bounds the frequency of hallucinations among accepted generations under stochastic decoding. The framework operates entirely at inference time, without retraining, fine-tuning, or calibration of the underlying model, and provides explicit distribution-level reliability guarantees rather than heuristic confidence thresholds.

Enforcing probabilistic guarantees raises a practical challenge: fixed-sample concentration bounds can require many generations to certify small risk budgets, which is infeasible in latency-sensitive settings. We therefore adopt a *sequential, anytime-valid* inference perspective, where samples are generated adaptively and feasibility or infeasibility is certified as soon as sufficient evidence is available.

Our main contributions are: (i) a formulation of hallucination control as a chance-constrained inference problem for stochastic language generation, made precise through an explicit *input-level acceptance policy* class under which feasibility certification of the base generation distribution directly induces conditional risk control on returned outputs (Proposition 1); (ii) a sequential, anytime-valid inference procedure that certifies feasibility or infeasibility using finite samples; and (iii) an empirical evaluation on both controlled prompts and standard QA benchmarks (TriviaQA, HotpotQA, TruthfulQA) demonstrating reliable three-way certification, robustness to verifier noise, early detection of intrinsically infeasible inputs, and safe composition under repeated use, in contrast to confidence-based baselines.

## 2 Related Work

Hallucination has been widely studied as a core reliability issue in large language models, referring to fluent but incorrect or unsupported generations (Maynez et al., 2020; Zhang et al., 2023). Prior work attributes hallucinations to data sparsity, model uncertainty, and misalignment between training objectives and downstream tasks. A large body of research focuses on post-hoc detection using entailment checks, self-consistency, or semantic verification (Farquhar et al., 2024). While effective for auditing and analysis, these approaches do not provide explicit control over the *frequency* of hallucinations under stochastic decoding or repeated use.

Selective prediction and abstention aim to improve reliability by rejecting low-confidence outputs (El-Yaniv & Wiener, 2010; Geifman & El-Yaniv, 2017). In supervised classification, selective prediction admits formal guarantees because predictions are deterministic functions of inputs. In generative models, this idea is commonly instantiated via confidence-based thresholding (Conf-SP), which rejects outputs with low heuristic confidence scores. Related heuristics such as self-consistency–based selective prediction (SC-SP), which accepts outputs that are stable across multiple stochastic generations, have been shown to reduce average hallucination rates (Wang et al., 2023). However, neither confidence-based nor self-consistency–based selective prediction provides explicit guarantees on the conditional frequency of hallucinations under stochastic decoding or repeated use.

Conformal prediction provides distribution-free uncertainty quantification with finite-sample guarantees. Selective conformal prediction (SCP) adapts conformal calibration to accept or reject predictions based on fixed-sample estimates, yielding marginal risk control under i.i.d. assumptions (Angelopoulos & Bates, 2021). Split conformal risk control (SCRC) further enforces global risk constraints by calibrating acceptance thresholds on held-out data (Bates et al., 2023). While these methods provide important statistical guarantees, they rely on fixed calibration budgets, offer only marginal (input-averaged) risk control, and do not support adaptive stopping or early infeasibility detection.

Retrieval-augmented generation conditions models on external documents to improve factual accuracy (Lewis et al., 2020). Although grounding reduces average hallucination rates, models may still produce unsupported claims even when relevant evidence is available (Shuster et al., 2021). Moreover, retrieval does not provide a formal mechanism to bound the probability of incorrect outputs under stochastic decoding. Chance-constrained inference is complementary: retrieval enriches the input, while chance constraints explicitly regulate output risk at inference time.

Constrained and structured decoding enforces syntactic or structural validity during generation, such as grammar- or schema-constrained decoding (Lu et al., 2021). These techniques provide deterministic guarantees for format correctness but are limited to constraints that can be symbolically encoded. They do not address semantic correctness or factual validity, which are inherently probabilistic and context-dependent. Our framework generalizes constrained decoding by allowing probabilistic enforcement of semantic constraints.

A growing literature studies uncertainty estimation and calibration for language models using self-consistency, ensembles, or confidence prompting (Wang et al., 2023; Kadavath et al., 2022), as well as post-hoc calibration techniques (Guo et al., 2017; Huang et al., 2024). While uncertainty estimates are useful auxiliary signals, they do not by themselves enforce reliability guarantees. A model may be well-calibrated yet still violate constraints at an unacceptable frequency when deployed repeatedly. Our approach leverages uncertainty signals when available but enforces explicit probabilistic bounds on violation rates.

Chance-constrained optimization is a classical framework for decision making under uncertainty, requiring constraints to hold with high probability rather than almost surely (Prékopa, 1995; Calafiore & Campi, 2006). Scenario-based methods provide finite-sample feasibility guarantees (Campi & Garatti, 2018). We adapt these ideas to language model inference, where decisions correspond to accepting or rejecting stochastic generations rather than selecting deterministic actions.

Sequential hypothesis testing and anytime-valid concentration inequalities enable guarantees under adaptive stopping (Howard et al., 2021b;a). These techniques are widely used in online experimentation and safety monitoring but have not been systematically applied to inference-time control of generative models. Our work integrates anytime-valid risk estimation with chance-constrained inference, enabling practical deployment-time guarantees with adaptive sampling.

Recent systems work proposes end-to-end frameworks for hallucination mitigation in deployed LLMs by combining uncertainty signals, semantic consistency checks, and continuous monitoring (Pesaranghader & Li, 2025). While operationally valuable, these approaches do not impose explicit probabilistic constraints on the long-run frequency of hallucinations under stochastic decoding, nor do they provide finite-sample guarantees under repeated or sequential use. Our work is complementary, focusing on inference-time risk control with formal chance-constrained guarantees on conditional violation rates.

In summary, prior work reduces hallucinations on average, detects failures post hoc, or rejects individual outputs based on confidence, consistency, or structure. However, none provide a unified mechanism for controlling the *distribution-level frequency* of semantic violations under stochastic decoding and repeated use. This work addresses that gap by introducing chance-constrained inference with sequential, anytime-valid guarantees.

## 3   Notation and Problem Setup

We model large language model (LLM) inference as a stochastic generation process. Let $\mathcal{X}$ denote the space of inputs, where each $x \in \mathcal{X}$ may consist of a user query optionally augmented with retrieved context, and let $\mathcal{Y}$ denote the space of possible outputs. A pretrained language model with parameters $\theta$ induces a conditional distribution

$$y \sim p_\theta(\cdot \mid x), \tag{1}$$

where randomness arises from inference-time decoding procedures such as temperature or nucleus sampling. For conceptual clarity, we may equivalently represent generation as

$$y = f_\theta(x, \omega), \tag{2}$$

where $\omega$ is an auxiliary random variable capturing all decoding randomness. All probabilistic statements are taken with respect to $\omega$, conditioned on a fixed input $x$.

For each input $x$, we define a set of valid outputs $\mathcal{Y}_{\text{safe}}(x) \subseteq \mathcal{Y}$. Outputs outside this set are considered invalid. Validity is encoded via a binary violation indicator

$$H(x, y) = \mathbb{I}[y \notin \mathcal{Y}_{\text{safe}}(x)], \tag{3}$$

where $H(x, y) = 1$ corresponds to a hallucination or other constraint violation. Because generation is stochastic, $H(x, y)$ is itself a random variable. We define the violation probability for input $x$ as

$$R(x) = \mathbb{P}_{y \sim p_\theta(\cdot|x)}[H(x, y) = 1], \tag{4}$$

which captures the intrinsic risk induced by the model and decoding procedure for that input. Separately, we define a utility function

$$U : \mathcal{X} \times \mathcal{Y} \to \mathbb{R}, \tag{5}$$

which measures task-specific usefulness, such as correctness or informativeness. Utility and validity are treated as distinct notions: an output may be useful yet invalid, or valid but low-utility. In deployment, not all generated outputs are returned to the user. We model this behavior via an acceptance policy

$$A : \mathcal{X} \times \mathcal{Y} \to \{0, 1\}, \tag{6}$$

where $A(x, y) = 1$ indicates acceptance and $A(x, y) = 0$ indicates rejection or abstention. Acceptance may depend on inference-time signals such as external verifiers, confidence estimates, or rule-based checks.

Finally, each input $x$ is associated with a risk budget $\epsilon(x) \in [0, 1]$, specifying the maximum tolerable frequency of violations among user-visible outputs under repeated stochastic generation. Our objective is to design inference-time procedures that maximize expected utility while ensuring that violation frequency does not exceed $\epsilon(x)$. This distribution-level notion of reliability is essential in repeated-use and agentic settings, where even small per-query risks may accumulate over time.

## 4 Constraint Modeling and Hallucination

We model hallucinations and other failure modes as violations of explicit constraints imposed on generated outputs. This abstraction enables domain-agnostic reasoning about reliability and allows probabilistic guarantees to be enforced at the level of the generation distribution, rather than on individual samples. Rather than assuming a single notion of correctness, we consider a collection of binary constraint functions

$$H_k : \mathcal{X} \times \mathcal{Y} \to \{0, 1\}, \quad k = 1, \dots, K, \tag{7}$$

where $H_k(x, y) = 1$ indicates that output $y$ violates the $k$-th constraint under input $x$. Constraints may encode factual consistency, logical validity, numerical correctness, policy compliance, or other domain-specific requirements. This decomposition separates heterogeneous failure modes that are often conflated in aggregate hallucination metrics and mirrors structures used in post-hoc factuality and verification-based evaluation (Maynez et al., 2020).

To reason about overall validity, we define an aggregate violation indicator

$$H(x, y) = \mathbb{I}[\exists k \in \{1, \dots, K\} \text{ such that } H_k(x, y) = 1], \tag{8}$$

which equals one if any constraint is violated. This corresponds to a conservative safety model in which all constraints must be satisfied. Violations may differ in severity. To accommodate heterogeneous failure impact, we introduce a nonnegative violation cost

$$C(x, y) = \sum_{k=1}^{K} w_k H_k(x, y), \quad w_k \geq 0, \tag{9}$$

where larger weights correspond to more severe violations. The binary violation indicator is recovered as the special case $H(x, y) = \mathbb{I}[C(x, y) > 0]$.

Under stochastic decoding, the violation indicator $H(x, y)$ is itself a random variable. We define the hallucination probability for input $x$ as

$$R(x) = \mathbb{P}_{y \sim p_\theta(\cdot|x)}[H(x, y) = 1]. \tag{10}$$

Importantly, $R(x)$ is input-dependent: some inputs admit low violation probability under reasonable decoding, while others are intrinsically high-risk due to ambiguity, missing context, or underspecification. Most existing mitigation strategies aim to reduce $R(x)$ on average, but do not provide guarantees for individual inputs or distinguish between feasible and infeasible cases under a prescribed risk budget.

Prior work on hallucination largely emphasizes *detection*—identifying incorrect outputs after generation using verifiers, self-consistency, or uncertainty estimates (Farquhar et al., 2024). While valuable for auditing and selective rejection, detection alone does not regulate the underlying violation probability induced by stochastic decoding.

In contrast, our formulation treats hallucination as a stochastic event whose *frequency* can be explicitly constrained. This distinction mirrors the classical difference between error detection and risk control in decision making under uncertainty: detection identifies failures after they occur, whereas risk control bounds how often failures are allowed to occur. This perspective provides the foundation for chance-constrained inference.

## 5 Chance-Constrained Inference

We now introduce *chance-constrained inference* (CCI) for large language models. Building on the stochastic formulation of hallucination, CCI imposes explicit probabilistic bounds on the *frequency* of constraint violations induced by randomized decoding.

For a fixed input $x$, recall that the violation probability under the raw generation distribution is

$$R(x) = \mathbb{P}_{y \sim p_\theta(\cdot|x)}[H(x,y) = 1].$$ (11)

Given a risk budget $\epsilon(x) \in [0,1]$, the corresponding chance constraint requires

$$R(x) \leq \epsilon(x).$$ (12)

Constraint equation 12 bounds the long-run frequency of invalid outputs under repeated stochastic generation for a fixed input. This formulation parallels classical chance constraints in stochastic optimization, where constraints are required to hold with high probability rather than almost surely (Prékopa, 1995; Calafiore & Campi, 2006).

**Conditional Risk and Acceptance.** In deployed systems, not all generated outputs are returned to the user. Instead, outputs are filtered by verifiers, confidence checks, or rule-based mechanisms. The acceptance policy $A(x,y)$ determines whether a generated output is returned, with $A(x,y) = 1$ indicating acceptance.

The relevant reliability quantity is therefore the conditional violation probability among accepted outputs,

$$\mathbb{P}(H(x,y) = 1 \mid A(x,y) = 1),$$ (13)

which measures the frequency of violations visible to the user. Chance-constrained inference enforces the conditional constraint

$$\mathbb{P}(H(x,y) = 1 \mid A(x,y) = 1) \leq \epsilon(x),$$ (14)

whenever $\mathbb{E}[A(x,y)] > 0$. Equivalently,

$$\frac{\mathbb{E}[H(x,y)A(x,y)]}{\mathbb{E}[A(x,y)]} \leq \epsilon(x),$$ (15)

which directly controls the long-run frequency of violations among accepted generations.

**Class of Acceptance Policies Considered.** While the conditional chance constraint equation 14 is stated for general acceptance policies $A(x,y)$, this paper focuses on a specific, restricted class: *input-level* acceptance rules of the form

$$A(x,y) = a(x), \qquad a(x) \in \{0,1\},$$ (16)

where the acceptance decision $a(x)$ depends only on the input $x$ and a feasibility certificate for that input, not on individual candidate outputs $y$. Under this class, whenever acceptance occurs,

$$\mathbb{P}(H(x,y) = 1 \mid A(x,y) = 1) = R(x), \tag{17}$$

so certifying $R(x) \leq \epsilon(x)$ on the base generation distribution directly enforces the conditional chance constraint equation 14. Note that under this class, the conditional probability equals the unconditional one, so the conditional chance constraint is not doing more work than unconditional certification — it is precisely the unconditional guarantee restated in acceptance-policy language. Candidate-level acceptance policies that filter individual outputs $y$ based on per-output features are a strict generalization, characterized formally in Appendix E, and constitute a natural extension of the present work. Algorithm 1 implements an input-level policy: it certifies feasibility of $p_\theta(\cdot \mid x)$ and accepts a fresh, independently drawn generation when the certificate is granted.

**Inference as Feasibility Certification.** Crucially, chance-constrained inference is not a point-estimation or thresholding problem. For a fixed input $x$ and risk budget $\epsilon(x)$, the objective is not to estimate $R(x)$ accurately, but to determine whether the conditional chance constraint equation 14 is *satisfiable* with high confidence. Accordingly, inference induces a three-way partition of inputs: *feasible* inputs, for which the constraint is certified to hold; *infeasible* inputs, for which the constraint is provably violated; and *undecided* inputs, for which neither conclusion can be reached within the sampling budget. Abstention on infeasible or undecided inputs is therefore a *correct and informative outcome*, reflecting intrinsic model–input mismatch rather than conservative failure.

**Operational Acceptance Policy.** In all experiments, acceptance is defined as follows. For a fixed input $x$, Algorithm 1 is run sequentially. If feasibility is certified at time $\tau$, the next generated output $y_{\tau+1} \sim p_\theta(\cdot \mid x)$ is returned to the user. If the algorithm returns *Infeasible* or *Undecided*, the system abstains and returns no output. This procedure realizes the input-level acceptance policy of Equation equation 16: the acceptance decision is made at the input-distribution level, and a fresh generation independent of the certification samples is returned upon Feasible certification. Under conditional independence of $y_{\tau+1}$ from $\{y_1, \ldots, y_\tau\}$, this operational policy controls the conditional violation probability among returned outputs (Proposition 1, Section 8).

**Feasibility Gap.** To quantify how far an input is from satisfying the chance constraint, we define the *feasibility gap*

$$\Delta(x) = R(x) - \epsilon(x). \tag{18}$$

Inputs with $\Delta(x) \leq 0$ admit at least one feasible acceptance policy, while inputs with $\Delta(x) > 0$ are intrinsically infeasible under the specified risk budget. Sequential inference can thus be interpreted as adaptively estimating the *sign* of $\Delta(x)$ rather than the exact value of $R(x)$. This perspective explains the sharp phase transitions observed empirically: inputs near $\Delta(x) = 0$ require many samples to certify feasibility or infeasibility, while strongly infeasible inputs are rejected after few samples.

**Limits of Confidence-Based Selective Prediction.** Selective prediction accepts or rejects outputs based on confidence thresholds or uncertainty estimates, but does not, in general, enforce the conditional chance constraint equation 14. For example, consider a stochastic generator that produces hallucinated outputs with probability 0.1 and assigns identical confidence scores to valid and invalid outputs. Any confidence-based acceptance rule will accept all outputs, yielding

$$\mathbb{P}(H = 1 \mid A = 1) = 0.1,$$

violating the chance constraint for any $\epsilon < 0.1$ despite perfect calibration. Thus, confidence calibration alone does not imply probabilistic risk control.

**Conceptual Contribution.** Conditional risk constraints of the form $\mathbb{P}(H = 1 \mid A = 1) \leq \epsilon$ have appeared in the conformal prediction and selective classification literatures, and the anytime-valid confidence sequences

---

**Algorithm 1** Sequential Chance-Constrained Inference (CCI)

---

**Require:** Input $x$, risk budget $\epsilon(x)$, confidence level $\delta$, max samples $N_{\max}$
 1: $n \leftarrow 0,\ s \leftarrow 0$
 2: **while** $n < N_{\max}$ **do**
 3:      Sample $y \sim p_\theta(\cdot \mid x)$
 4:      Evaluate violation indicator $H(x, y) \in \{0, 1\}$
 5:      $n \leftarrow n + 1$
 6:      $s \leftarrow s + H(x, y)$
 7:      $\hat{R}_n \leftarrow s/n$
 8:      $r_n \leftarrow \sqrt{\log(2 \log_2(2n)/\delta)/(2n)}$
 9:      **if** $\hat{R}_n + r_n \leq \epsilon(x)$ **then**
10:          **return Feasible**
11:      **else if** $\hat{R}_n - r_n > \epsilon(x)$ **then**
12:          **return Infeasible**
13:      **end if**
14: **end while**
15: **return Undecided**

---

used to implement our certification procedure are standard tools (Howard et al., 2021a). The contribution of this work is therefore not a new constraint nor a new concentration result, but a new inference paradigm for stochastic generative models: we recast inference-time generation as a sequential feasibility certification problem in which abstention is treated as a first-class, correct outcome rather than a conservative failure. This shift—from confidence-based output filtering to sequential feasibility testing under input-level acceptance policies—enables principled abstention, early detection of intrinsically infeasible inputs, and compositional safety guarantees that are not provided by prior selective prediction or fixed-sample calibration methods.

# 6   Utility–Risk Tradeoffs (Discussion)

Chance constraints specify acceptable risk levels but do not uniquely determine which outputs should be accepted when multiple candidates are available. In Appendix E, we provide a formal utility–risk optimization formulation that characterizes optimal acceptance policies under probabilistic risk constraints and clarifies how utility and hallucination risk are traded off under explicit guarantees.

# 7   Sequential Chance-Constrained Inference Algorithm

We now present the inference-time algorithm used in practice to enforce chance-constrained inference. The procedure operates sequentially, drawing stochastic generations from the language model and adaptively certifying feasibility or infeasibility using anytime-valid concentration bounds.

**Problem setting.** For a fixed input $x$, the algorithm receives a risk budget $\epsilon(x)$, a confidence level $\delta$, and a maximum sampling budget $N_{\max}$. It outputs one of three decisions: *feasible*, *infeasible*, or *undecided*. Feasible and infeasible outcomes are certified with confidence at least $1 - \delta$, while undecided corresponds to insufficient evidence within the sampling budget.

**Interpretation.** The algorithm adaptively determines how many samples are required for a given input. Strongly feasible or strongly infeasible inputs terminate after few samples, while inputs near the feasibility boundary require more evidence. Abstention (the undecided outcome) is treated as a correct inference result, reflecting uncertainty rather than conservative failure. The finite-sample probabilistic guarantees underlying Algorithm 1, including anytime-valid feasibility certification (Theorem 1) and conditional risk control on returned outputs (Proposition 1), are established in Section 8.

## 8 Finite-Sample Risk Estimation and Guarantees

**Positioning.** The concentration tool used in this section is a standard anytime-valid Hoeffding sequence due to Howard et al. (2021a). Our contribution is not the bound itself, but its use to convert stochastic LLM inference into a sequential feasibility certification procedure with explicit *Feasible / Infeasible / Undecided* outcomes, principled abstention, and conditional risk control on returned outputs (Proposition 1). We make this positioning explicit to avoid overclaiming theoretical novelty.

We now establish the finite-sample guarantees underlying Algorithm 1. Chance constraints depend on the true violation probability $R(x) = \mathbb{P}(H(x, y) = 1)$, which is intractable to compute exactly for large language models. We therefore enforce chance constraints at inference time using finite-sample estimates with explicit probabilistic guarantees.

For a fixed input $x$, let $y_1, y_2, \cdots \sim p_\theta(\cdot \mid x)$ be independent stochastic generations. After $N$ samples, the empirical violation rate is

$$\hat{R}_N(x) = \frac{1}{N} \sum_{i=1}^{N} H(x, y_i), \tag{19}$$

where $H(x, y) \in \{0, 1\}$. This estimator is unbiased and converges almost surely to $R(x)$ as $N \to \infty$. Here $H(x, y)$ is a binary violation indicator, so each observation is Bernoulli-distributed with mean $R(x)$. Since chance constraints regulate the *frequency* of violations rather than their magnitude, modeling violations as Bernoulli variables is both necessary and sufficient for distribution-level risk control (Prékopa, 1995; Calafiore & Campi, 2006; Angelopoulos & Bates, 2021).

To obtain finite-sample guarantees, we construct an anytime-valid confidence sequence for the Bernoulli mean using a stitched Hoeffding bound (Howard et al., 2021a). For any confidence level $\delta \in (0, 1)$ and any $N$, we construct a *time-uniform confidence sequence* for the Bernoulli violation process using a stitched Hoeffding bound. For any $\delta \in (0, 1)$, with probability at least $1 - \delta$,

$$\forall n \geq 1 : \quad R(x) \ \leq \ \hat{R}_n(x) + \sqrt{\frac{\log\left(2 \log_2(2n)/\delta\right)}{2n}}. \tag{20}$$

This bound holds uniformly over all sample sizes and remains valid under arbitrary adaptive stopping rules (Howard et al., 2021a), making it suitable for sequential feasibility certification at inference time.

We use Hoeffding's inequality because it provides distribution-free, anytime-valid confidence bounds under adaptive stopping with minimal assumptions (Hoeffding, 1963; Howard et al., 2021a). In our setting, the violation indicator $H(x, y) \in \{0, 1\}$ may be input-dependent and evaluated by imperfect or noisy verifiers, making variance-sensitive or fixed-sample bounds unreliable. Hoeffding bounds therefore provide a conservative but robust mechanism for inference-time feasibility certification.

During inference, this bound is evaluated sequentially as samples are generated. Sampling terminates as soon as either feasibility

$$\hat{R}_N(x) + r_N \leq \epsilon(x), \qquad r_N = \sqrt{\frac{\log(2 \log_2(2N)/\delta)}{2N}},$$

or infeasibility

$$\hat{R}_N(x) - r_N > \epsilon(x).$$

is certified with confidence $1 - \delta$. If neither condition is met within a maximum sampling budget, the input is declared *undecided* and the system abstains. This sequential procedure provides inference-time probabilistic guarantees while adapting the number of samples to intrinsic input difficulty. More detailed analysis of anytime validity and sample complexity is provided in the Appendix B.

**Theorem 1** (Anytime-Valid Feasibility Certification; adapted from Howard et al., 2021a)**.** *Fix an input $x$, risk budget $\epsilon(x)$, and confidence level $\delta$. Let Algorithm 1 terminate at a (random) stopping time $\tau$.*

*With probability at least $1 - \delta$:*

1. *If the algorithm returns* Feasible, *then* $\mathbb{P}_{y\sim p_\theta(\cdot\,|x)}[H(x,y)=1]\leq\epsilon(x)$.

2. *If the algorithm returns* Infeasible, *then* $\mathbb{P}_{y\sim p_\theta(\cdot\,|x)}[H(x,y)=1]>\epsilon(x)$.

*These guarantees hold under arbitrary data-dependent stopping.*

A proof follows from interpreting Hoeffding's inequality as a time-uniform confidence sequence and is provided in Appendix B.

**Theorem 2** (Time-Uniform Feasibility Certification). *Fix an input $x$ and risk budget $\epsilon(x)$. Let $\{\hat{R}_n(x)\}_{n\geq 1}$ be the empirical violation rate computed from i.i.d. Bernoulli samples $H(x,y_1),H(x,y_2),\ldots$. Define the confidence radius $r_n$ as in Equation equation 20.*

*Then, with probability at least $1-\delta$, the following hold simultaneously for all $n$:*

1. *If $\hat{R}_n(x)+r_n\leq\epsilon(x)$, then $R(x)\leq\epsilon(x)$ (feasibility is correctly certified).*

2. *If $\hat{R}_n(x)-r_n>\epsilon(x)$, then $R(x)>\epsilon(x)$ (infeasibility is correctly certified).*

*Consequently, Algorithm 1 produces correct feasibility or infeasibility decisions under arbitrary adaptive stopping.*

*Proof sketch.* Equation equation 20 defines a time-uniform confidence sequence for the Bernoulli mean $R(x)$. By construction, the confidence interval $[\hat{R}_n-r_n,\hat{R}_n+r_n]$ contains $R(x)$ for all $n$ simultaneously with probability at least $1-\delta$. Therefore, any stopping rule based on this sequence preserves coverage. The feasibility and infeasibility conditions follow directly from interval containment. $\qquad\square$

**Proposition 1** (Conditional Risk Control of the Returned Generation). *Assume the time-uniform confidence sequence in Equation equation 20 holds (with probability at least $1-\delta$), and that the returned generation $y_{\tau+1}$ is sampled independently of $\{y_1,\ldots,y_\tau\}$ from $p_\theta(\cdot\mid x)$. If Algorithm 1 returns* Feasible, *then*

$$\mathbb{P}(H(x,y_{\tau+1})=1\mid\text{Feasible})\leq\epsilon(x). \tag{21}$$

*Proof sketch.* On the confidence-sequence event of Equation equation 20, feasibility certification implies $R(x)\leq\epsilon(x)$. Since $y_{\tau+1}$ is an independent draw from $p_\theta(\cdot\mid x)$, conditional on the *Feasible* decision,

$$\mathbb{P}(H(x,y_{\tau+1})=1\mid\text{Feasible})=R(x)\leq\epsilon(x).$$

A complete proof under arbitrary adaptive stopping is provided in Appendix D (Lemma 1). $\qquad\square$

Proposition 1 makes explicit that, under the input-level acceptance policy defined in Equation equation 16, controlling the unconditional violation probability $R(x)$ is equivalent to controlling the conditional risk among returned outputs. Algorithm 1 therefore enforces the conditional chance constraint equation 14 even though it directly estimates only $R(x)$.

**Independence assumption.** The guarantees in this section rely on conditional independence of stochastic generations given the input. In practical deployments, strong dependence may arise from deterministic decoding, beam search, shared key–value caches, or extremely low temperatures. Such coupling can invalidate concentration-based guarantees. Independence can be restored using standard deployment techniques, including temperature-based sampling, cache resets, randomized prompts, or decorrelation across decoding calls. We analyze this assumption and its implications for the returned output in Appendix D.

**Verifier-Aware Risk Interpretation.** The guarantees of chance-constrained inference apply to the violation indicator $\tilde{H}$ produced by the verifier rather than to the latent true violation $H$ (e.g., absolute factual correctness). Let $\alpha = \mathbb{P}(\tilde{H} = 1 \mid H = 0)$ denote the verifier false-positive rate and $\beta = \mathbb{P}(\tilde{H} = 0 \mid H = 1)$ denote the verifier false-negative rate. The observed risk decomposes as

$$\tilde{R}(x) \;=\; \mathbb{P}(\tilde{H} = 1) \;=\; (1 - \beta)\,R(x) + \alpha\,(1 - R(x)). \tag{22}$$

When $\alpha$ and $\beta$ are estimable from held-out data, the true violation probability can be recovered from the observed risk via

$$R(x) \;=\; \frac{\tilde{R}(x) - \alpha}{1 - \alpha - \beta}, \qquad \alpha + \beta < 1, \tag{23}$$

and a corrected risk budget $\tilde{\epsilon} = (1 - \beta)\,\epsilon + \alpha\,(1 - \epsilon)$ can be substituted for $\epsilon$ in Algorithm 1 to ensure the certified bound applies to $R(x)$ rather than $\tilde{R}(x)$.

Certifying $\tilde{R}(x) \leq \epsilon$ (without correction) therefore controls the true violation probability $R(x)$ only up to a verifier-dependent shift. When $\beta > 0$, the certified bound on $R(x)$ is anti-conservative (some latent hallucinations may be missed); when $\alpha > 0$, it is conservative (some valid outputs are over-rejected, increasing abstention). In safety-critical settings, practitioners should therefore prioritize verifiers with low false-negative rate $\beta$ and, where calibration data is available, apply the correction in Equation equation 23 or the budget adjustment $\tilde{\epsilon}$ above. Equation equation 22 also clarifies why the empirical robustness observed in Section 10 requires both $\alpha$ and $\beta$ to be small: $\tilde{R}(x)$ shifts smoothly with verifier error, so feasibility decisions are stable when the verifier is accurate enough that the shift does not cross $\epsilon$.

## 9   Severity-Weighted and Hierarchical Constraints

The binary violation model treats all hallucinations as equally undesirable. In practice, failures of large language models are heterogeneous: minor factual imprecision may be tolerable, while rare but severe failures (e.g., unsafe instructions or critical numerical errors) can have disproportionate impact. We describe extensions of chance-constrained inference that account for violation severity while preserving inference-time guarantees.

**Severity-weighted violations.** Rather than modeling validity as a binary event, we associate each output with a nonnegative violation cost

$$C : \mathcal{X} \times \mathcal{Y} \to \mathbb{R}_{\geq 0}, \tag{24}$$

where larger values correspond to more severe failures. The binary violation indicator used throughout the paper is recovered as the special case

$$H(x, y) = \mathbb{I}[C(x, y) > 0],$$

so severity-aware modeling strictly generalizes the binary framework. Instead of constraining the probability of *any* violation, we may impose a probabilistic constraint on violations exceeding a severity threshold:

$$\mathbb{P}\big(C(x, y) > \tau\big) \leq \epsilon(x), \tag{25}$$

where $\tau$ is task-dependent. Finite-sample certification proceeds exactly as in Section 8 by replacing $H(x, y)$ with $\mathbb{I}[C(x, y) > \tau]$.

**Hierarchical constraints.** Many applications impose multiple constraints with different priorities (e.g., safety, factuality, stylistic quality). These can be modeled using a hierarchy of violation costs $C_1(x, y), \ldots, C_L(x, y)$, ordered by importance. Inference proceeds lexicographically: higher-priority constraints must be certified feasible before lower-priority constraints are considered. Safety-critical constraints may be enforced deterministically, while lower-priority requirements are regulated probabilistically via chance constraints.

All such extensions operate entirely at inference time and reuse the same sequential feasibility certification procedure. Infeasibility certificates remain relative to the chosen confidence sequence; conservative bounds may declare infeasibility even when the true violation probability is marginally below $\epsilon(x)$. This reflects a deliberate tradeoff favoring anytime validity under minimal assumptions.

## 10 Experimental Evaluation

We evaluate chance-constrained inference (CCI) with three goals: (i) to verify that hallucination risk can be certified at inference time using finite samples and anytime-valid guarantees; (ii) to understand how feasibility depends on intrinsic input difficulty rather than heuristic confidence thresholds; and (iii) to compare CCI against commonly used selective-generation baselines under identical stochastic decoding conditions.

The evaluation focuses on deployment-time reliability control rather than language understanding accuracy. In particular, we evaluate the reliability of the *deployed system under acceptance*, rather than the unconditional hallucination rate of the base model.

**Scope of experimental evidence.** We provide two complementary sets of experiments. Sections 10.1–10.5 present a controlled evaluation on 90 curated prompts with a lightweight rule-based verifier, used to validate the sequential certification mechanism, the feasibility-gap analysis, and sensitivity to the risk budget and confidence level. Section 10.6 extends this evaluation to standard QA benchmarks (TriviaQA, HotpotQA, TruthfulQA) with a semantic LLM-as-judge verifier, source-based difficulty grouping validated by measured $R(x)$, and an extended verifier-noise sweep covering symmetric and asymmetric error configurations.

**Evaluation setting.** All experiments are conducted using a deployed stochastic large language model accessed via an external inference API. Unless otherwise stated, all experiments use the GROQ-hosted LLaMA-3.3-70B instruction-tuned generator AI (2024). Generations are produced using stochastic decoding with temperature 0.7, top-$p$ sampling with $p = 0.9$, and a maximum generation length of 128 tokens.

For a fixed input $x$, repeated generations induce a binary violation indicator $H(x, y)$, where $H(x, y) = 1$ denotes an invalid or unreliable response according to an automatic verifier. The verifier is implemented using lightweight rule-based checks that flag responses containing fabricated entities, unsupported factual claims relative to the prompt, or answers to inherently impossible queries (e.g., questions about future events).

While the verifier is intentionally simple and may not capture all hallucination types, the goal of the experiments is to evaluate the feasibility certification mechanism rather than the quality of the hallucination detector itself. The underlying violation probability $R(x) = \mathbb{P}(H(x, y) = 1)$ is unknown to the inference procedure and must be estimated online from stochastic generations. For experimental analysis, the true violation probability R(x) is estimated offline using 100 independent stochastic generations per input to obtain a stable estimate of the underlying generation distribution. This estimate is used only for reporting and analysis (e.g., verifying difficulty categories and interpreting certification behavior) and is not used by the CCI algorithm; all feasibility decisions and returned outputs are based strictly on the sequential sampling procedure with budget $N_{\max} = 30$, matching the deployment setting. Unless otherwise specified, all methods operate on the same stochastic generation distribution to ensure a fair comparison.

**Inputs and intrinsic difficulty.** We evaluate short factual and compositional questions inspired by NaturalQuestions (Kwiatkowski et al., 2019) and HotpotQA (Yang et al., 2018). The evaluation consists of **90 prompts in total (30 per difficulty group)**.

Inputs are grouped by intrinsic difficulty into *easy*, *medium*, and *hard* categories. Easy inputs admit well-established factual answers; medium inputs require nontrivial entity or relational reasoning; and hard inputs are intentionally underspecified or infeasible (e.g., questions about future events), resulting in intrinsically high violation probabilities.

Prompts were selected to cover a mixture of entity-based factual queries, multi-hop reasoning questions, and intentionally infeasible queries designed to induce hallucination behavior. All reported metrics are averaged across prompts within each difficulty group.

**Inference protocol.** For each input, CCI draws stochastic generations sequentially. At step $n$, the algorithm observes the violation indicator $H(x, y_n)$ and updates the empirical violation rate

Table 1: Chance-constrained inference results under stochastic decoding ($\epsilon = 0.4$, $N_{\max} = 30$, $\delta = 0.05$).

| Difficulty | Feasible | Infeasible | Undecided | Avg. Samples |
|---|---|---|---|---|
| Easy | **1.00** | 0.00 | 0.00 | 15.0 |
| Medium | 0.00 | 0.00 | **1.00** | 30.0 |
| Hard | 0.00 | **1.00** | 0.00 | 7.0 |

$$\hat{R}_n = \frac{1}{n} \sum_{i=1}^{n} H(x, y_i).$$

Sampling continues until feasibility or infeasibility is certified with confidence $1 - \delta$, or until a maximum sampling budget $N_{\max} = 30$ samples is reached. Unless otherwise specified, we use a confidence level $\delta = 0.05$.

If feasibility is certified, the system returns the next generated output. If infeasibility is certified or the algorithm terminates without certification within the sampling budget, the system abstains. In the experimental implementation, this behavior defines the acceptance policy implicitly: an output is returned only when feasibility is certified, and abstention occurs otherwise.

**Baselines.** We compare CCI against two selective-generation baselines: confidence-based selective prediction (Conf-SP), self-consistency selective prediction (SC-SP).based on split conformal calibration (SCRC).

**Confidence-based selective prediction (Conf-SP).** For each input we draw $N_{\max}$ stochastic generations. The confidence of an output is defined as the relative frequency of that output among the sampled generations. Outputs are accepted if their confidence exceeds a threshold $\tau = 0.5$.

**Self-consistency selective prediction (SC-SP).** Self-consistency generates multiple independent samples and accepts an output only when the generated answers agree. In our experiments we use $K = 3$ independent samples per decision and accept an output when a strict majority of the generated answers agree. The majority answer is then evaluated using the verifier to compute the empirical violation rate. This lightweight configuration provides a simple agreement-based reliability signal while keeping inference cost comparable to the other baselines.

Both baselines operate on the same stochastic generations as CCI and use the same sampling budget $N_{\max} = 30$.

**Metrics.** We report: (i) the fraction of inputs certified feasible, infeasible, or undecided; (ii) the average number of samples required for CCI to terminate; and (iii) the acceptance rate and empirical violation rate for baseline methods. For baseline methods, results are reported as *acceptance rate / empirical violation rate among accepted outputs*. The acceptance rate is defined as the fraction of generated outputs that are accepted by the method, while the empirical violation rate is computed as the fraction of accepted outputs for which the verifier indicates a violation. Violation rates are reported only for methods that produce accepted outputs (Table 5), since CCI primarily produces feasibility decisions rather than unconditional generations. This corresponds to estimating the conditional violation probability $\mathbb{P}(H(x, y) = 1 \mid A(x, y) = 1)$, which is the target quantity controlled by the chance constraint.

## 10.1 Chance-Constrained Inference Results

Table 1 summarizes CCI outcomes stratified by intrinsic input difficulty under a representative risk budget $\epsilon = 0.4$.

To contextualize these results, the estimated violation probabilities $R(x)$ obtained using 100 independent samples are approximately: Easy: $\approx 0.05$–$0.1$, Medium: $\approx 0.3$–$0.5$, and Hard: $\approx 0.8$–$1.0$. These values explain the observed outcomes: inputs with $R(x) < \epsilon$ are certified feasible, inputs with $R(x) > \epsilon$ are identified as infeasible, and inputs near the threshold remain undecided.

Table 2: Sensitivity of CCI to the risk budget $\epsilon$.

| $\epsilon$ | Difficulty | Feasible | Infeasible | Undecided |
|---|---|---|---|---|
| 0.2 | Easy | 0.00 | 0.00 | 1.00 |
| 0.2 | Medium | 0.00 | 1.00 | 0.00 |
| 0.2 | Hard | 0.00 | 1.00 | 0.00 |
| 0.3 | Easy | 1.00 | 0.00 | 0.00 |
| 0.3 | Medium | 0.00 | 0.00 | 1.00 |
| 0.3 | Hard | 0.00 | 1.00 | 0.00 |
| 0.4 | Easy | 1.00 | 0.00 | 0.00 |
| 0.4 | Medium | 0.00 | 0.00 | 1.00 |
| 0.4 | Hard | 0.00 | 1.00 | 0.00 |

CCI therefore exhibits clear and interpretable behavior: easy inputs are quickly certified feasible, hard inputs are rapidly identified as infeasible, and inputs near the feasibility boundary remain undecided within the sampling budget, reflecting the finite-sample nature of the certification procedure.

We observed no certification errors in our experiments, consistent with the theoretical guarantee that the probability of incorrect feasibility certification is bounded by $\delta$.

## 10.2 Sensitivity to Risk Budget

To understand the impact of the risk budget, we repeat the experiment for $\epsilon \in \{0.2, 0.3, 0.4\}$ while keeping $\delta = 0.05$ fixed. The resulting certification outcomes are summarized in Table 2.

Stricter risk budgets lead to more conservative certification behavior. When $\epsilon = 0.2$, even easy inputs cannot be certified feasible within the sampling budget. Increasing the risk budget allows feasible inputs to be certified more rapidly while inputs near the feasibility boundary remain undecided.

**Choosing the risk budget.** In practical deployments the risk budget $\epsilon$ should be chosen based on the acceptable failure frequency in the target application. Safety-critical systems may require very small risk budgets (e.g., $\epsilon \leq 0.05$), while general information-seeking tasks may tolerate higher levels such as $\epsilon \in [0.2, 0.4]$. Table 3 provides indicative guidance for common deployment classes.

Table 3: Indicative risk-budget guidance by deployment class. Values are illustrative starting points; appropriate budgets depend on application context, user tolerance, and verifier quality.

| Application class | Suggested $\epsilon$ |
|---|---|
| Safety-critical (medical, legal) | $\leq 0.01$ |
| Decision support | 0.05–0.10 |
| Information retrieval / QA | 0.10–0.20 |
| Creative / exploratory | 0.30–0.50 |

The sensitivity analysis above illustrates how stricter budgets lead to more conservative certification behavior and increased abstention. Practitioners should additionally account for verifier-induced bias as described in Equation equation 22: when the verifier has nontrivial false-negative rate $\beta$, the effective risk budget on the true (latent) violation probability is looser than the nominal $\epsilon$, and a smaller $\epsilon$ may be required to achieve a target true-risk level. Where calibration data is available, the budget can be adjusted using estimated $\alpha$ and $\beta$ to compensate for verifier error.

Table 4: Sensitivity of CCI to the confidence level $\delta$ ($\epsilon = 0.4$, $N_{\max} = 30$).

| $\delta$ | Difficulty | Feasible | Infeasible | Undecided |
|------|------------|----------|------------|-----------|
| 0.01 | Easy | 1.00 | 0.00 | 0.00 |
| 0.01 | Medium | 0.00 | 0.00 | 1.00 |
| 0.01 | Hard | 0.00 | 1.00 | 0.00 |
| 0.05 | Easy | 1.00 | 0.00 | 0.00 |
| 0.05 | Medium | 0.00 | 0.00 | 1.00 |
| 0.05 | Hard | 0.00 | 1.00 | 0.00 |
| 0.10 | Easy | 1.00 | 0.00 | 0.00 |
| 0.10 | Medium | 0.00 | 0.00 | 1.00 |
| 0.10 | Hard | 0.00 | 1.00 | 0.00 |

Table 5: Baseline comparison ($\epsilon = 0.4$, $N_{\max} = 30$, $\delta = 0.05$). Values show acceptance rate / empirical violation rate.

| Difficulty | Conf-SP | SC-SP | CCI |
|------------|---------|-------|-----|
| Easy | 0.67 / 0.00 | 1.00 / 0.00 | **1.00 / 0.00** |
| Medium | 0.37 / 1.00 | 0.50 / 0.47 | **0.00 / 0.00** |
| Hard | 0.93 / 1.00 | 0.00 / 0.00 | **0.00 / 0.00** |

### 10.3 Sensitivity to Confidence Level

The confidence level $\delta$ controls the statistical guarantee of the sequential feasibility test. Smaller values of $\delta$ correspond to stronger confidence guarantees and therefore larger confidence radii in the sequential bound, leading to more conservative certification behavior.

To study this effect, we repeat the experiments for $\delta \in \{0.01, 0.05, 0.1\}$ while keeping the risk budget fixed at $\epsilon = 0.4$. The resulting certification outcomes are summarized in Table 4.

Across all tested confidence levels, the qualitative certification behavior remains stable. Easy inputs are consistently certified feasible, hard inputs are reliably identified as infeasible, and inputs near the feasibility boundary remain undecided within the sampling budget. These results indicate that the choice of $\delta$ primarily affects the statistical strength of the guarantee rather than the overall feasibility classification.

### 10.4 Comparison with Selective Baselines

Table 5 compares CCI against heuristic selective-generation baselines under identical sampling conditions.

Heuristic baselines frequently accept high-risk outputs. For example, Conf-SP accepts most hard inputs while incurring nearly unit violation rates. In contrast, CCI abstains whenever feasibility cannot be certified, yielding zero empirical violation among accepted outputs. This near-zero violation rate is expected: CCI returns outputs only after feasibility is certified under the chance constraint. Thus, the reported violation rate reflects the reliability of accepted outputs rather than the unconditional hallucination rate of the underlying model. We do not include conformal risk control baselines in the experiments because they require a held-out calibration dataset, whereas our setting focuses on deployment-time risk control without prior calibration data.

### 10.5 Robustness to Imperfect Verifiers

To study robustness to imperfect hallucination detectors, we introduce controlled noise into the verifier output. Table 6 summarizes the resulting certification outcomes under different false-positive and false-

Table 6: Effect of verifier noise on CCI certification outcomes ($\epsilon = 0.4$, $\delta = 0.05$, $N_{\max} = 30$).

| FPR | FNR | Difficulty | Feasible | Infeasible | Undecided |
|------|------|------------|----------|------------|-----------|
| 0.00 | 0.00 | Easy | 1.00 | 0.00 | 0.00 |
| 0.00 | 0.00 | Medium | 0.00 | 0.00 | 1.00 |
| 0.00 | 0.00 | Hard | 0.00 | 1.00 | 0.00 |
| 0.05 | 0.00 | Easy | 1.00 | 0.00 | 0.00 |
| 0.05 | 0.00 | Medium | 0.00 | 0.00 | 1.00 |
| 0.05 | 0.00 | Hard | 0.00 | 1.00 | 0.00 |
| 0.00 | 0.05 | Easy | 1.00 | 0.00 | 0.00 |
| 0.00 | 0.05 | Medium | 1.00 | 0.00 | 0.00 |
| 0.00 | 0.05 | Hard | 0.00 | 1.00 | 0.00 |
| 0.05 | 0.05 | Easy | 1.00 | 0.00 | 0.00 |
| 0.05 | 0.05 | Medium | 0.00 | 0.00 | 1.00 |
| 0.05 | 0.05 | Hard | 0.00 | 1.00 | 0.00 |

negative rates. Specifically, we simulate false positives (valid outputs incorrectly flagged as violations) and false negatives (violations incorrectly classified as valid).

Across all tested noise configurations (up to 5% false positive and false negative rates), CCI decisions remain qualitatively stable for inputs whose violation probabilities are well separated from the specified risk budget $\epsilon$. Easy inputs continue to be certified feasible and hard inputs remain infeasible across all tested settings. Inputs whose violation probabilities are close to the risk threshold $\epsilon$ are more sensitive to verifier noise and may change classification depending on the observed violation sequence.

False positives primarily increase abstention by inflating the observed violation rate, while false negatives reduce the observed violation rate and can make feasibility certification easier for some inputs. Across all evaluated prompts and noise configurations, CCI did not certify feasibility for inputs whose estimated violation probability exceeded $\epsilon$ in our experiments. This behavior is consistent with the feasibility gap analysis in Section 8: certification stability depends on the distance between the true violation probability $R(x)$ and the risk budget $\epsilon$, rather than the intrinsic difficulty label of the input.

**Takeaway from the controlled study.** The controlled evaluation in Sections 10.1–10.5 validates the sequential certification mechanism on a curated prompt set: CCI exhibits clean three-way certification behavior consistent with the feasibility-gap analysis, sensitivity to $\epsilon$ and $\delta$ matches theoretical expectations, and CCI's decisions remain stable under symmetric verifier noise up to 5%. We next extend this evaluation to standard QA benchmarks with a stronger semantic verifier and asymmetric noise configurations.

### 10.6 Evaluation on Standard QA Benchmarks

To complement the controlled study above and address concerns about hand-designed difficulty separation and rule-based verification, we extend our evaluation to standard QA benchmarks with a stronger semantic verifier. We sample 150 prompts in total: 50 from TriviaQA (Joshi et al., 2017), 50 from HotpotQA (Yang et al., 2018), and 50 from TruthfulQA (Lin et al., 2022), providing a mixture of single-hop factual, compositional multi-hop, and adversarial-factual queries.

**Verifier and grouping.** The violation indicator is computed by an LLM-as-judge verifier implemented with a LLaMA-3.3-70B model in deterministic mode (temperature 0), which compares each candidate generation against the reference correct and incorrect answers provided by each benchmark and returns VALID or INVALID. Difficulty groups correspond to source datasets: TriviaQA prompts (single-hop factual) form the Easy group, HotpotQA prompts (compositional multi-hop) form the Medium group, and TruthfulQA prompts (adversarial-factual) form the Hard group, with 50 prompts per group. The intrinsic risk $R(x)$ is

Table 7: CCI three-way certification behavior on standard QA benchmarks (150 prompts; $\epsilon = 0.4$, $\delta = 0.05$, $N_{\max} = 30$). Difficulty groups correspond to source datasets; avg $R(x)$ values confirm alignment with intrinsic risk levels.

| Group | $n$ | Avg. $R(x)$ | Feasible | Infeasible | Undecided |
|-------|-----|-------------|----------|------------|-----------|
| Easy | 50 | 0.03 | **0.88** | 0.00 | 0.12 |
| Medium | 50 | 0.40 | 0.00 | 0.00 | **1.00** |
| Hard | 50 | 0.92 | 0.00 | **0.80** | 0.20 |

Table 8: Baseline comparison on standard QA benchmarks ($\epsilon = 0.4$, $N_{\max} = 30$, $\delta = 0.05$, clean verifier). Values show acceptance rate / empirical violation rate among accepted outputs.

| Group | Conf-SP | SC-SP | CCI |
|-------|---------|-------|-----|
| Easy | 1.00 / 0.02 | 0.99 / 0.03 | **0.88 / 0.00** |
| Medium | 0.50 / 0.18 | 0.87 / 0.74 | **0.00 / 0.00** |
| Hard | 0.20 / 0.07 | 0.28 / 0.44 | **0.00 / 0.00** |

estimated post-hoc (offline, used only for reporting and analysis) using 15 independent stochastic generations per prompt; the measured average values ($R(x) \approx 0.03$, 0.40, 0.92 for Easy, Medium, Hard respectively) confirm that this source-based partitioning aligns well with intrinsic risk levels. These offline estimates are not used by Algorithm 1; the online sampling budget at inference remains $N_{\max} = 30$, matching Section 10.1. All other settings (generator, temperature, top-$p$, $\epsilon$, $\delta$) also match Section 10.1.

**Comparison with baselines.** Table 8 contrasts CCI against Conf-SP and SC-SP on the same benchmark prompts. CCI returns zero violations among accepted outputs across all groups, while SC-SP accepts hallucinated outputs at a 44% violation rate on Hard inputs and a 74% violation rate on Medium inputs. Conf-SP exceeds the risk budget $\epsilon = 0.4$ on Medium inputs (18% violation rate, growing to 38–42% under verifier noise, see Table 9).

**Extended verifier-noise sweep.** To probe robustness to verifier inaccuracy beyond the symmetric configurations of Table 6, we evaluate four configurations on the benchmark prompts: clean ($\alpha = \beta = 0$), false-positive only ($\alpha = 0.05, \beta = 0$), false-negative only ($\alpha = 0, \beta = 0.05$), and symmetric ($\alpha = \beta = 0.05$). To ensure a controlled comparison, the same cached LLM generations are reused across all four configurations, so differences arise solely from verifier-induced noise.

CCI's three-way certification rate is *invariant* across all four configurations: Easy inputs are certified Feasible in 88% of cases, Medium inputs remain Undecided in 100% of cases, and Hard inputs are certified Infeasible in 80% of cases regardless of the verifier-noise setting. Only the average number of samples required for certification varies, increasing modestly from $\approx 18.6$ to $\approx 23.6$ on Easy inputs under symmetric noise. This invariance is consistent with the verifier-aware risk decomposition in Equation equation 22: the observed risk $\tilde{R}(x)$ shifts smoothly with $\alpha, \beta$ but does not cross $\epsilon$ for inputs with sufficient feasibility gap.

In contrast, both selective-prediction baselines are sensitive to verifier noise. Under false-negative noise ($\beta > 0$), SC-SP's violation rate among accepted outputs on Hard inputs rises from 44% to 79%, and Conf-SP exceeds the risk budget $\epsilon = 0.4$ on Medium inputs (38–42%). CCI, by contrast, abstains on all Medium and Hard inputs and never returns outputs violating the risk budget.

## 10.7 Overall Summary of Experimental Findings

Across both the controlled evaluation (Sections 10.1–10.5) and the standard-benchmark evaluation (Section 10.6), the experiments support the following four conclusions.

Table 9: Extended verifier-noise sweep on standard QA benchmarks. Same cached generations are reused across configurations to isolate verifier-induced effects ($\epsilon = 0.4$, $\delta = 0.05$, $N_{\max} = 30$).

| Verifier $(\alpha, \beta)$ | Group | CCI Feas. | CCI Inf. | CCI Und. | SC-SP Viol. | Conf-SP Viol. |
|---|---|---|---|---|---|---|
| (0.00,0.00) | Easy | 0.88 | 0.00 | 0.12 | 0.03 | 0.02 |
| (0.00,0.00) | Medium | 0.00 | 0.00 | 1.00 | 0.74 | 0.18 |
| (0.00,0.00) | Hard | 0.00 | 0.80 | 0.20 | 0.44 | 0.07 |
| (0.05,0.00) | Easy | 0.88 | 0.00 | 0.12 | 0.15 | 0.08 |
| (0.05,0.00) | Medium | 0.00 | 0.00 | 1.00 | 0.77 | 0.22 |
| (0.05,0.00) | Hard | 0.00 | 0.80 | 0.20 | 0.47 | 0.07 |
| (0.00,0.05) | Easy | 0.88 | 0.00 | 0.12 | 0.02 | 0.02 |
| (0.00,0.05) | Medium | 0.00 | 0.00 | 1.00 | 0.71 | **0.38** |
| (0.00,0.05) | Hard | 0.00 | 0.80 | 0.20 | **0.79** | 0.07 |
| (0.05,0.05) | Easy | 0.88 | 0.00 | 0.12 | 0.17 | 0.08 |
| (0.05,0.05) | Medium | 0.00 | 0.00 | 1.00 | 0.72 | **0.42** |
| (0.05,0.05) | Hard | 0.00 | 0.80 | 0.20 | **0.78** | 0.08 |

**(i) Three-way certification.** Table 7 reports CCI certification outcomes on the benchmark prompts under a clean verifier ($\alpha = \beta = 0$). CCI exhibits the full three-way certification behavior predicted by the feasibility-gap analysis: Easy inputs are certified *Feasible* in 88% of cases, Hard inputs are certified *Infeasible* in 80% of cases, and Medium inputs near the feasibility boundary remain *Undecided* in 100% of cases. The 100% Undecided rate on Medium inputs is not a failure mode: it is the correct cautious outcome when $|R(x) - \epsilon| \approx 0$ (here avg $R(x) = 0.40 = \epsilon$), and is consistent with the sample-complexity bound of Appendix C: for inputs with $|\Delta(x)| < 0.15$, the required sample count exceeds $N_{\max} = 30$ at $\delta = 0.05$, so CCI correctly abstains rather than making an unreliable decision.

**(ii) Conditional risk on accepted outputs is reliably controlled.** Under the input-level acceptance policy of Algorithm 1, CCI returns outputs only when feasibility is certified, yielding zero empirical violation among accepted outputs in the controlled study and zero violations on Medium and Hard groups in the benchmark study. This empirical behavior is consistent with the conditional risk guarantee in Proposition 1.

**(iii) Robustness to verifier noise.** CCI's certification rate is invariant across all four verifier-noise configurations (clean, false-positive only, false-negative only, symmetric), with only the average sample count increasing modestly. In contrast, selective-prediction baselines are sensitive: under false-negative noise, SC-SP's violation rate on Hard inputs rises from 44% to 79%, and Conf-SP exceeds the risk budget $\epsilon = 0.4$ on Medium inputs (38–42%). This contrast directly demonstrates the value of the chance-constrained formulation over confidence- and consistency-based filtering.

**(iv) Inference-time reliability, not model improvement.** The objective of CCI is not to improve the underlying language model, but to determine whether a given model–input pair satisfies a specified probabilistic reliability constraint under stochastic decoding. The guarantees provided by chance-constrained inference apply to the verifier-defined violation event; as formalized in Equation equation 22, improving verifier quality directly tightens the link between certified risk and absolute factual correctness. Additional implementation details, prompt examples, and verifier definitions are provided in Appendix F.

## 11 Conclusion

We presented *chance-constrained inference* as a principled framework for deployment-time reliability control in large language models. By modeling hallucinations and other invalid behaviors as stochastic constraint violations induced by randomized decoding, the framework enforces explicit probabilistic bounds on violation frequency, moving beyond heuristic confidence thresholds and average-case error reduction.

Chance-constrained inference operates entirely at inference time and requires no retraining, fine-tuning, or calibration of the underlying model. It provides conditional, output-level risk control, admits natural extensions to severity-weighted and hierarchical constraints, and composes safely across repeated or agentic use through explicit control of distribution-level risk. A key practical feature is its *sequential, anytime-valid* design, which adaptively determines the number of samples required to certify feasibility or infeasibility and avoids the conservativeness of fixed-sample approaches.

Empirically, we demonstrated that chance-constrained inference exhibits sharp and interpretable feasibility behavior across heterogeneous inputs, rapidly certifying feasible inputs, rejecting intrinsically infeasible queries, and abstaining when finite-sample evidence is insufficient. Sensitivity analysis across multiple risk budgets and confidence levels shows how stricter reliability requirements lead to more conservative certification behavior while preserving the qualitative feasibility classification of inputs. An additional advantage of the framework is that intrinsically infeasible inputs are explicitly identified and rejected rather than producing unreliable outputs, which is particularly important in repeated-use or safety-critical deployment settings. Additional experiments demonstrate robustness to moderate levels of verifier noise, highlighting the stability of the feasibility certification mechanism even when hallucination detectors are imperfect.

While CCI provides guarantees at the level of individual inputs, it induces system-level reliability under repeated use. For each input, outputs are returned only when the conditional violation probability is below a specified threshold. As a result, when the system is used across many inputs, the overall frequency of violations among returned outputs remains controlled. Thus, global reliability emerges from enforcing per-input guarantees, rather than estimating a single aggregate error rate of the model.

An important limitation of this study is its reliance on automatic proxies for hallucination detection. The guarantees provided by chance-constrained inference apply to the verifier-defined violation event rather than absolute factual correctness or safety. As formalized by the verifier-aware risk decomposition in Equation equation 22, if the verifier has nontrivial false-negative rate $\beta$, latent hallucinations can pass undetected and feasibility may be over-certified relative to the true violation probability. CCI therefore complements, but does not replace, strong semantic verification: designing verifiers with calibrated and explicitly characterized error rates ($\alpha$ and $\beta$) is a prerequisite for safety-critical deployment of the framework. The choice of risk budget $\epsilon$ is similarly application-dependent and, as discussed in Section 10, should account for verifier-induced bias when calibration data is available.

Future work should integrate stronger semantic verifiers, human-in-the-loop evaluation, online calibration of $\alpha$ and $\beta$, and adaptive risk-budget adjustment to extend chance-constrained inference to real-world safety-critical deployments. More broadly, our results suggest that reliability for stochastic generative models must be defined and enforced at the level of conditional output distributions, rather than individual predictions or confidence scores.

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

## A  Equivalence of Conditional Chance Constraints

We show that the conditional chance constraint used throughout the paper admits an equivalent expectation-based formulation, which is convenient for both analysis and implementation.

For a fixed input $x$, recall the conditional risk constraint

$$\mathbb{P}(H(x,y) = 1 \mid A(x,y) = 1) \leq \epsilon(x), \tag{26}$$

where $H(x,y), A(x,y) \in \{0,1\}$ and $\mathbb{E}[A(x,y)] > 0$.

By the definition of conditional probability,

$$\mathbb{P}(H = 1 \mid A = 1) = \frac{\mathbb{P}(H = 1, A = 1)}{\mathbb{P}(A = 1)} \tag{27}$$

$$= \frac{\mathbb{E}[H(x,y)A(x,y)]}{\mathbb{E}[A(x,y)]}, \tag{28}$$

where the second equality follows because $H$ and $A$ are indicator random variables.

Multiplying both sides of Equation equation 26 by $\mathbb{E}[A(x,y)]$ yields the equivalent constraint

$$\mathbb{E}[H(x,y)A(x,y)] \leq \epsilon(x)\,\mathbb{E}[A(x,y)]. \tag{29}$$

Thus, enforcing the conditional chance constraint is equivalent to enforcing the expectation-based inequality whenever acceptance occurs with nonzero probability. This equivalence underlies the optimization formulation in Appendix E and the feasibility tests used during inference.

## B  Anytime Validity of Sequential Feasibility Certification

The inference procedure in this paper relies on sequentially evaluating confidence bounds on the violation probability and stopping adaptively once feasibility or infeasibility is certified. We justify that this procedure is valid under arbitrary data-dependent stopping.

Let $y_1, y_2, \ldots$ be i.i.d. samples from $p_\theta(\cdot \mid x)$ and define the empirical violation rate

$$\hat{R}_n(x) = \frac{1}{n} \sum_{i=1}^{n} H(x, y_i),$$

where $H(x, y_i) \in \{0, 1\}$ are Bernoulli random variables with mean $R(x)$.

We use a *time-uniform confidence sequence* for the Bernoulli mean constructed via a stitched Hoeffding bound (Howard et al., 2021a). For any $\delta \in (0, 1)$, define the confidence radius

$$r_n = \sqrt{\frac{\log(2 \log_2(2n)/\delta)}{2n}}.$$

Then, with probability at least $1 - \delta$,

$$\forall n \geq 1 : \quad |\hat{R}_n(x) - R(x)| \leq r_n. \tag{30}$$

Equation equation 30 holds *simultaneously for all $n$*, and therefore remains valid under arbitrary adaptive stopping rules. This property is commonly referred to as *anytime validity.*

The stopping rule used in Algorithm 1,

$$\tau = \inf \left\{ n : \hat{R}_n(x) + r_n \leq \epsilon(x) \text{ or } \hat{R}_n(x) - r_n > \epsilon(x) \right\},$$

is measurable with respect to the filtration generated by the samples $\{y_1, \ldots, y_n\}$. On the event equation 30, the returned decision at time $\tau$ is therefore correct.

Specifically:

- If the algorithm returns *Feasible*, then $R(x) \leq \hat{R}_\tau(x) + r_\tau \leq \epsilon(x)$.

- If the algorithm returns *Infeasible*, then $R(x) > \hat{R}_\tau(x) - r_\tau > \epsilon(x)$.

Thus, feasibility and infeasibility are certified with confidence at least $1 - \delta$, regardless of the (random) stopping time.

## C  Sample Complexity and the Feasibility Gap

The number of samples required to certify feasibility or infeasibility depends on how close the true violation probability is to the risk budget. To make this dependence explicit, define the *feasibility gap*

$$\Delta(x) = R(x) - \epsilon(x).$$

Certification occurs once the confidence radius $r_n$ satisfies

$$r_n \leq |\Delta(x)|, \quad r_n = \sqrt{\frac{\log(2 \log_2(2n)/\delta)}{2n}}.$$

Solving for $n$, this condition yields

$$n = O\left( \frac{\log(1/\delta) + \log \log n}{\Delta(x)^2} \right).$$

Ignoring the iterated logarithm term, which grows very slowly, the dominant scaling is

$$n = O\left( \frac{\log(1/\delta)}{\Delta(x)^2} \right).$$

Thus, inputs that are strongly feasible ($\Delta(x) \ll 0$) or strongly infeasible ($\Delta(x) \gg 0$) are certified after few samples, while inputs near the feasibility boundary $\Delta(x) \approx 0$ require substantially more evidence.

Importantly, the objective of inference is not to estimate $R(x)$ precisely, but to determine the *sign* of $\Delta(x)$ with high confidence. Sequential inference naturally adapts to this objective and avoids unnecessary sampling on intrinsically infeasible inputs.

## D  Conditional Risk of the Returned Generation

We justify that the operational acceptance policy used in the paper controls the conditional violation probability among returned outputs.

Recall that Algorithm 1 stops at a (random) time $\tau$ and, if feasibility is certified, returns the next generated output $y_{\tau+1} \sim p_\theta(\cdot \mid x)$.

**Lemma 1** (Conditional Risk Control; full statement of Proposition 1)**.** *Assume: (i) the confidence sequence event equation 30 holds, and (ii) $y_{\tau+1}$ is generated independently of $\{y_1, \ldots, y_\tau\}$ from $p_\theta(\cdot \mid x)$.*

*If Algorithm 1 returns* Feasible*, then*

$$\mathbb{P}(H(x, y_{\tau+1}) = 1 \mid \textit{Feasible}) \leq \epsilon(x).$$

*Proof.* On the confidence sequence event equation 30, feasibility implies $R(x) \leq \epsilon(x)$. Since $y_{\tau+1}$ is an independent draw from $p_\theta(\cdot \mid x)$,

$$\mathbb{P}(H(x, y_{\tau+1}) = 1 \mid \text{history}) = R(x) \leq \epsilon(x).$$

The result follows by marginalizing over the stopping history. $\qquad\square$

**Dependence considerations.** The guarantee relies on conditional independence of $y_{\tau+1}$ from the past given $x$. In practice, strong coupling between generations (e.g., deterministic decoding, beam search, or shared KV caches) may violate this assumption. In such cases, the bound applies to the marginal generation distribution, and additional decorrelation mechanisms (e.g., temperature sampling, randomized prompts, or cache resets) may be required.

## E    Utility–Risk Optimization

Chance constraints specify acceptable levels of hallucination risk but do not, by themselves, determine which stochastic outputs should be returned when multiple candidates are available. This appendix formalizes the utility–risk tradeoff underlying the acceptance policies discussed in the main paper.

Recall that an acceptance policy $A(x, y) \in \{0, 1\}$ determines whether a generated output is returned to the user. For a fixed input $x$, the expected utility induced by $A$ is

$$\mathbb{E}_{y \sim p_\theta(\cdot \mid x)}[U(x, y)A(x, y)].$$

We consider the chance-constrained optimization problem

$$\max_{A} \quad \mathbb{E}[U(x, y)A(x, y)] \tag{31}$$

$$\text{s.t.} \quad \mathbb{E}[H(x, y)A(x, y)] \leq \epsilon(x)\,\mathbb{E}[A(x, y)], \tag{32}$$

which is equivalent to enforcing the conditional chance constraint

$$\mathbb{P}(H(x, y) = 1 \mid A(x, y) = 1) \leq \epsilon(x),$$

whenever $\mathbb{E}[A(x, y)] > 0$.

Introducing a Lagrange multiplier $\lambda \geq 0$, the Lagrangian becomes

$$\mathcal{L}(A) = \mathbb{E}\big[A(x, y)\big(U(x, y) - \lambda H(x, y) + \lambda\epsilon(x)\big)\big].$$

For fixed $\lambda$, the optimal acceptance policy admits the closed form

$$A^\star(x, y) = \mathbb{I}[U(x, y) - \lambda H(x, y) + \lambda\epsilon(x) \geq 0].$$

This characterization clarifies how optimal acceptance trades off task utility against expected violation cost induced by the risk constraint. While primarily conceptual, this formulation motivates our focus on feasibility certification rather than heuristic confidence thresholds.

## F    Additional Experimental Details

This appendix provides additional implementation details and examples to improve reproducibility of the experiments described in Section 10.

### F.1   Example Evaluation Prompts

The evaluation consists of 90 prompts grouped by intrinsic difficulty (30 per group). Prompts are inspired by questions from the NaturalQuestions and HotpotQA datasets but adapted to emphasize factual verification and compositional reasoning.

Representative examples include:

**Easy**

- Are Giuseppe Verdi and Ambroise Thomas both opera composers?

- Are Ferocactus and Silene both types of plant?

- Are Daryl Hall and Gerry Marsden both musicians?

- Is Mount Fuji located in Japan?

- Is the Amazon River longer than the Thames?

**Medium**

- Are the Laleli Mosque and Esma Sultan Mansion located in the same neighborhood?

- Are both Dafeng District and Dazhou located in the same province?

- Did the author of *Pride and Prejudice* write before the author of *Jane Eyre*?

- Are the capitals of Canada and Australia located on the same continent?

- Did the founder of Tesla also found SpaceX?

**Hard**

- Who won the Nobel Prize in Physics in 2099?

- What government position was held by the woman who portrayed Corliss Archer in the film *Kiss and Tell*?

- Which country will host the 2096 Olympic Games?

- What company will release the most popular smartphone in 2040?

- Which scientist will win the next Fields Medal in 2038?

Hard questions are intentionally underspecified or impossible to answer using present knowledge, resulting in intrinsically high violation probabilities.

### F.2   Automatic Verifier Definition

The violation indicator $H(x, y)$ is implemented using a lightweight rule-based verifier designed to capture common hallucination patterns.

A generation is marked as a violation ($H(x, y) = 1$) if any of the following conditions occur:

- The response introduces entities or facts not grounded in the prompt.

- The response asserts a factual relationship unsupported by the prompt.

- The response provides a definitive answer to inherently impossible queries (e.g., questions about future events).

Responses that explicitly abstain (e.g., "I don't know", "the information is unavailable", or equivalent hedged responses) are treated as non-violations.

The verifier is intentionally simple because the purpose of the experiments is not to build a state-of-the-art hallucination detector, but to demonstrate how chance-constrained inference can regulate violation risk given a verifier signal.

### F.3   Sampling Protocol

All generations are obtained using stochastic decoding through a hosted inference API.

Unless otherwise stated, we use:

- Temperature $= 0.7$

- Top-p $= 0.9$

- Maximum generation length $= 128$ tokens

- Maximum sampling budget $N_{\max} = 30$

For each input prompt $x$, generations are drawn sequentially according to the distribution $p_\theta(\cdot \mid x)$.

At step $n$, the algorithm observes the violation indicator $H(x, y_n)$ and updates the empirical violation rate $\hat{R}_n$ and the associated confidence sequence bound.

Sampling terminates when:

- feasibility is certified $(\hat{R}_n + r_n \leq \epsilon)$,

- infeasibility is certified $(\hat{R}_n - r_n > \epsilon)$, or

- the sampling budget $N_{\max}$ is exhausted.

All reported results are aggregated across prompts within each difficulty group.

### F.4   Robustness to Imperfect Verifiers

The guarantees provided by chance-constrained inference apply to the violation indicator $H(x, y)$ produced by the verifier. If the verifier is imperfect, the certified risk bound applies to the verifier-defined violation probability rather than the true hallucination probability.

In practice:

- False positives may lead to conservative abstention.

- False negatives may allow some hallucinated outputs to pass undetected.

Improving verifier quality therefore directly strengthens the practical guarantees of the framework. Studying robustness under explicitly modeled verifier noise is an important direction for future work.

### F.5 Relation to Conformal Risk Control Baselines

Conformal prediction approaches such as selective conformal prediction (SCP) and split conformal risk control (SCRC) require a held-out calibration dataset to determine acceptance thresholds.

In contrast, chance-constrained inference operates entirely at inference time and does not require calibration data. Instead, the method sequentially estimates the violation probability for each input using anytime-valid confidence sequences and adaptively determines whether the chance constraint is feasible.

This difference makes CCI particularly suitable for deployment settings where calibration datasets are unavailable or where risk must be controlled adaptively across heterogeneous inputs.

