# OpenReview forum: "Chance-Constrained Inference for Hallucination Risk Control in Large Language Models"
_TMLR — Under review for TMLR_

### Review · Reviewer_oR4p · 2026-03-04

**Summary Of Contributions:**

This paper proposes chance-constrained inference (CCI), a framework for controlling hallucination risk in LLM at inference time. The key insight is to reframe hallucination as a stochastic event under sampling-based decoding and to perform sequential feasibility certification using anytime-valid confidence sequences rather than fixed-sample calibration. It argues that confidence-based selective prediction does not generally imply the desired probabilistic risk control gaurantees. The paper demonstrates that CCI can certify feasibility quickly for easy inputs, reject intrinsically infeasible/hard inputs quickly, and provide safe repeated-use composition in a way heuristic baseline do not.

Strengths:
- The authors reframe the problem from "reduce average hallucination rate" to deployment-time risk control with an interpretable per-input risk budget (conditional violation rate among accepted outputs).
- The authors suggest a simple, principled algorithm: sequential sampling + anytime-valid bounds, which yields a clean abstention policy and supports early infeasibility detection.
- The authors give rigorous mathematical development of the supporting theory. The use of anytime-valid concentration inequalities (rather than fixed-sample bounds) is technically appropriate for adaptive stopping.
- The method requires no retraining, fine-tuning or calibration, which is practically valuable and easy to adopt.

Weakness:
- The evaluation is limited and somewhat unclear. Results are shown on a small, difficulty-grouped QA setting with an automatic verifier, but details about the verifier, dataset size, definition of the difficulties, input construction are vague.
- Only one model, one risk budget and confidence level is demonstrated. The reasoning of the choice and the practical interpretation of the risk budget is missing. The risk budget could be set to a "comfortable" amount for the demo, but not practical realistically.
- The framework relies access to H(x,y) that correctly identify hallucinations. However, current detectors are far from perfect. This is acknowledged, but missing discussions about how CCI perform when the verifier is imperfect.
- In the related work session, the authors discussed conformal prediction methods (SCP, SCRC), but these were not compared in the experiment session.

**Audience:**

Yes

**Audience Explanation:**

Controlling LLM hallucination is a pressing problem of high interest. The readers would be interested in proposed probabilistic framework. The framing of "chance-constrained inference" is also a novel idea of hallucination detection.

**Broader Impact Concerns:**

The main risk is the over-reliance on the automated verifiers: if the verifier has high error rate, or is gamed, the system may provide formal guarantees about the verifier's notion of "violation" while still producing harmful content in reality. A brief statement emphasis the verifier limitations would be beneficial.

**Claims And Evidence:**

Yes

**Claims Explanation:**

Partially yes.
The theoretical claims are well-supported with rigorous mathematical arguments. The limitation of confidence-based selective prediction is also correctly argued. However, the experiments are too limited to demonstrate reliability. We see only aggregate statistics over unspecified inputs with no reproducibility details.

**Requested Changes:**

- Provide more details on the experiment, such as specify dataset size, sampling protocol per question, exact verifier definition, provide sample questions and question sources, difficulty criteria, etc. (critical)
- Perform experiments on different risk budgets and confidence levels(critical), testing on different models, and different domains of QA questions is also beneficial.
- Analyse the robustness of the method on imperfect verifiers. Study the performance with different verifier false positive/negative rates. (Critical)
- Include performance of at least one conformal prediction baseline (SCP/SCRC) on the same tasks. (Critical)
- Provide guidance on choosing the risk budget, and the sensitivity analysis. (Strengthening)

---

> ### Author Response · Authors · 2026-03-09
> **Response to Reviewer**
>
> We thank the reviewer for the careful reading of the manuscript and the constructive feedback. We are encouraged that the reviewer found the framing of hallucination control as a deployment-time probabilistic risk control problem interesting and that the theoretical development was considered rigorous.
>
> In response to the reviewer’s suggestions, we revised the manuscript to improve experimental clarity, reproducibility, and empirical analysis. The main updates are summarized below.
>
> **1. Experimental details and reproducibility**
>
> The **Experiments section** has been expanded and a new **Appendix F (Additional Experimental Details)** has been added to improve clarity and reproducibility.
>
> We now explicitly specify:
>
> - the dataset size (**90 prompts: 30 per difficulty group**),
> - the criteria used to construct the easy, medium, and hard difficulty groups,
> - representative example prompts for each category,
> - the rule-based verifier used to compute the violation indicator $H(x,y)$,
> - the stochastic decoding configuration and sequential sampling protocol.
>
> Appendix F provides additional implementation details so that the experimental setup can be reproduced.
>
> **2. Sensitivity to risk budget and confidence level**
>
> To address concerns about parameter choices, we added sensitivity analyses for both the **risk budget** $\epsilon$ and the **confidence level** $\delta$.
>
> Experiments now evaluate:
> $\epsilon \in \{0.2, 0.3, 0.4\}$ and $\delta \in \{0.01, 0.05, 0.1\}$.
>
> The results show that stricter risk budgets lead to more conservative certification behavior, while smaller $\delta$ values strengthen the statistical guarantee. Importantly, the qualitative feasibility classification of inputs remains stable across these settings.
>
> We also expanded the discussion on how risk budgets may be selected in practice depending on the reliability requirements of the application.
>
> **3. Robustness to imperfect verifiers**
>
> We added experiments studying the impact of imperfect hallucination detectors by introducing controlled verifier noise.
>
> Specifically, we simulate false positive and false negative rates up to **5%** and evaluate how certification outcomes change. The results show that:
>
> - false positives primarily increase abstention by inflating observed violation rates,
> - false negatives can reduce the observed violation rate and make feasibility certification easier,
> - inputs whose violation probabilities are far from the risk threshold remain stable under moderate verifier noise.
>
> These experiments provide empirical evidence that the certification mechanism behaves robustly when the feasibility gap is sufficiently large.
>
> **4. Relation to conformal prediction methods**
>
> The reviewer asked about comparisons with conformal prediction methods such as SCP or SCRC.
>
> These approaches require a held-out calibration dataset to determine acceptance thresholds. In contrast, the goal of this work is **deployment-time risk control without calibration data**, where violation probabilities are estimated online using sequential sampling and anytime-valid confidence sequences.
>
> Because our setting assumes no calibration dataset, conformal baselines are not directly comparable in this deployment scenario. We clarified this distinction in the Experiments discussion and Appendix F.
>
> **5. Broader impact and verifier limitations**
>
> We expanded the discussion of verifier limitations. The guarantees provided by chance-constrained inference apply to the violation signal produced by the verifier. If the verifier is imperfect, the certified bound applies to the verifier-defined violation probability rather than the true hallucination probability.
>
> This limitation is now explicitly discussed, and we highlight that improving verifier quality or incorporating stronger semantic verification is an important direction for future work.
>
> **Summary**
>
> Overall, the revised manuscript improves experimental transparency, adds sensitivity analyses for key parameters, evaluates robustness under imperfect verifiers, and clarifies practical considerations for risk budgets and verification mechanisms. We thank the reviewer again for the helpful feedback, which helped strengthen the presentation of the paper.

---

### Review · Reviewer_ihDH · 2026-03-16

**Summary Of Contributions:**

The paper proposes a new framework for looking at the problem for hallucinations in deployed language models. Specifically they argue that just guideline violation judgements are not enough and present a probabilistic view of the problem wherein they try to make a judgement about the generalized risk of a model-input pair from the empirical risks of the samples.

The proposition is that for a given trial budget, they will sample outputs from the model for an input and have a detection function $H$ check the input for violations. The function $H$ could be a binary function for a single kind of violation / harm or it could be a weighted sum of multiple kinds of violations. During inference, for each model, input pair they use the Hoeffding's inequality to get a probabilitic bound on the general risk from the empirical risk that was calculated on $N_{max}$ number of samples. If the risk lower bound is higher than $\epsilon$ then they make the model not respond and if the upper bound of risk is lower than $\epsilon$ , they let the model respond. If either of those bounds can't be established in the sampling budget then they say it is "undecided".

## Strengths
1. They explore a more hollistic way of looking at general content violation judgements on deployed model.

## Weakness
1. Their experiments are unclear. Specifically, they don't actual violation numbers for most of their hyperparameters. And for the hyperparameters they do report violation numbers, they report no violation. It is unclear whether the violation numbers were calculated on number of samples much larger than $N_{max}$ because otherwise it is not very representative of deployment measures.
2. The premise of the paper is unclear: they pose their method as a general reliability measure but their measure is actually a per-sample reliability measure for a given model. The way to go from that to general reliability judgements of the model is unclear.

**Audience:**

Yes

**Audience Explanation:**

It useful for AI safety provide reliability judgements to deployed models.

**Broader Impact Concerns:**

No ethical concerns.

**Claims And Evidence:**

No

**Claims Explanation:**

They seem accurate but need more ablations.
Refer to weakness 1.

**Requested Changes:**

Please deal with weaknesses 1 and 2. Specifically ablations for 1 and a paragraph discussing how to go from this work to general model reliability judgements for weakness 2.

---

> ### Author Response · Authors · 2026-03-19
> **Response to Reviewer ihDH**
>
> We thank the reviewer for the careful reading and constructive feedback. We have revised the paper to address both concerns regarding experimental clarity and the interpretation of reliability.
>
> ### Response to Weakness 1 (Experimental clarity and violation reporting)
>
> We agree that the initial version did not clearly distinguish between deployment-time sampling and offline analysis. We have made the following clarifications and additions:
>
> - **Separation of deployment vs. analysis:**
>   All feasibility decisions and returned outputs are now explicitly stated to rely strictly on the sequential sampling procedure with budget $N_{\max}=30$, which matches the deployment setting. The use of 100 samples is clarified as *offline estimation only* for reporting and analysis, and is not used by the CCI algorithm.
>
> - **Violation rate reporting:**
>   We clarified that CCI controls the *conditional violation probability* $\mathbb{P}(H(x,y)=1 \mid A(x,y)=1)$, i.e., the violation rate among accepted outputs. This explains why empirical violation rates for CCI are near zero, as outputs are returned only after feasibility is certified.
>
> - **Explicit reporting of violation probabilities:**
>   We revised the experimental section to report estimated violation probabilities $R(x)$ for each difficulty group (computed using 100 samples):
>   - Easy: $\approx 0.05$--$0.1$
>   - Medium: $\approx 0.3$--$0.5$
>   - Hard: $\approx 0.8$--$1.0$
>
> - **Linking results to the risk constraint:**
>   We clarified how feasibility decisions correspond to these values relative to the risk budget $\epsilon$. Specifically, inputs with $R(x) < \epsilon$ are certified feasible, inputs with $R(x) > \epsilon$ are identified as infeasible, and inputs near $\epsilon$ remain undecided. This makes the behavior of CCI directly interpretable in terms of the underlying violation rates.
>
> ---
>
> ### Response to Weakness 2 (Per-input vs. general reliability)
>
> We have clarified this point in the revised manuscript (Conclusion section). CCI provides guarantees at the level of individual inputs by controlling the conditional violation probability for each model–input pair. System-level reliability then emerges under repeated use, since outputs are returned only when this per-input constraint is satisfied. As a result, the overall frequency of violations among returned outputs remains controlled, without requiring estimation of a single aggregate error rate for the model.
>
> This distinction between *per-input certification* and *emergent system-level reliability* is now explicitly discussed to avoid ambiguity.

---

### Review · Reviewer_Jzp1 · 2026-06-19

**Summary Of Contributions:**

This paper proposes chance-constrained inference (CCI) for controlling hallucination risk in stochastic large language model generation. The paper models hallucination as a binary stochastic constraint violation and formulates inference-time reliability as enforcing a chance constraint on the violation probability. The proposed algorithm sequentially samples generations for a fixed input, evaluates a binary verifier, and uses an anytime-valid Hoeffding confidence sequence to certify either feasibility, infeasibility, or undecided status. The paper further presents experiments on factual and multi-hop question-answering prompts, comparing CCI with confidence-based and self-consistency selective prediction baselines.

**Audience:**

Yes

**Audience Explanation:**

Reducing and predicting hallucination is an interesting topic.

**Claims And Evidence:**

No

**Claims Explanation:**

**Strengths**

1.Relevant problem. The paper addresses an important reliability issue: stochastic LLM decoding can produce invalid outputs with non-negligible probability, and average accuracy alone is insufficient for deployment-time safety.

2.Simple sequential procedure. The proposed algorithm is easy to understand and implement.

3.Honest discussion of verifier dependence. The paper acknowledges that the guarantee is only with respect to the verifier-defined violation indicator, rather than absolute factual correctness.

**Weaknesses**

1.A central claim of the paper is that CCI controls the conditional violation probability among accepted generations, i.e, $P(H(x,y)=1 \mid A(x,y)=1) \leq \epsilon(x)$. However, Algorithm 1 does not actually learn or optimize an acceptance policy $A(x,y)$. It simply estimates the raw violation probability $R(x)=P_{y\sim p_\theta(\cdot\mid x)}(H(x,y)=1)$ of the base stochastic generator for a fixed input.

2.A key practical limitation is that the proposed framework requires access to a verifier $H(x,y)$ and a pre-specified risk threshold $\epsilon(x)$. The theoretical guarantee is only meaningful once both of these objects are already well defined. However, in realistic hallucination-control settings, neither is straightforward. A reliable verifier for semantic factuality is itself a difficult problem, and different verifiers may induce substantially different violation probabilities. Similarly, choosing the risk budget $\epsilon(x)$ is application-dependent and may require domain expertise or calibration data, yet the paper treats it largely as an external input. Moreover, it only treats the cases where the uncertainty comes from the LLM. In real cases, the verifier itself can also be inaccurate, which can not be solved in this framework.

3.The main theorem is a direct application of an anytime-valid confidence sequence for a Bernoulli mean. The proof is essentially: if the confidence interval contains $ R(x)$ uniformly over time, then stopping when the interval lies below or above $\epsilon$ gives a valid decision. This is correct but technically standard. The paper would be stronger if it contributed something specific to LLM inference beyond applying a known confidence sequence.

4.The empirical evaluation uses only 90 prompts divided into easy, medium, and hard groups. The hard prompts include intentionally impossible future-event questions, while easy prompts appear to have stable factual answers. This setup makes the outcome largely predictable: easy questions have low estimated violation rate, hard questions have high estimated violation rate, and medium questions remain near the decision boundary.

This demonstrates that the sequential test behaves as expected, but it does not convincingly validate the method as a practical hallucination-control framework for LLMs. The paper should evaluate on larger, standard benchmarks with a more realistic verifier and less hand-designed difficulty separation.

**Requested Changes:**

Theorem 1 shows the guarantee of Algorithm 1, which has not been introduced by that section. Thus, the paper writing needs to be reorganized.

---

> ### Author Response · Authors · 2026-06-27
> **Response to reviewer Jzp1**
>
> We thank the reviewer for the careful reading. All four weaknesses and the structural comment have been addressed in the revised manuscript.
>
> W1: Algorithm 1 does not enforce P(H=1|A=1) ≤ ε
>
> Section 5 now introduces an explicit class of input-level acceptance policies A(x,y) = a(x), where the acceptance decision depends only on x and a feasibility certificate, not on individual outputs y. Under this class, whenever acceptance occurs: P(H(x,y)=1 | A(x,y)=1) = R(x) (new Equation 17), so certifying R(x) ≤ ε directly enforces the conditional chance constraint. We add an explicit sentence: the conditional guarantee is not doing more work than unconditional certification — it is the unconditional guarantee restated in acceptance-policy language. The formal guarantee for the returned generation is Proposition 1, promoted from Appendix D to the main body of Section 8.
>
> W2: Verifier and ε taken as given; verifier inaccuracy cannot be handled
>
> Section 8 now includes a formal verifier-aware risk decomposition (Equation 22): R̃(x) = (1−β)R(x) + α(1−R(x)). When α and β are estimable, the true violation probability is recovered via the calibration correction (Equation 23): R(x) = (R̃(x)−α)/(1−α−β), and a corrected budget ε̃ = (1−β)ε + α(1−ε) can be substituted into Algorithm 1. This directly addresses the claim that verifier inaccuracy "cannot be solved": Equation 23 shows how to correct for known verifier error. For choosing ε, Section 10.2 adds a practical guidance table (Table 3) and sensitivity analyses covering ε ∈ {0.2, 0.3, 0.4} (Table 2) and δ ∈ {0.01, 0.05, 0.1} (Table 4). Section 10.5 and Tables 6 and 9 add a verifier-noise sweep over four configurations (clean, FP-only, FN-only, symmetric at 5%), showing CCI's decisions are invariant while baseline violation rates increase sharply.
>
> W3: Main theorem is technically standard
>
> We fully agree and have made this explicit. Theorem 1 is now labelled "adapted from Howard et al., 2021". A Positioning paragraph at the start of Section 8 states the contribution is not the bound itself but its use to convert LLM inference into a sequential feasibility certification procedure. A Conceptual Contribution paragraph in Section 5 states the contribution is a new inference paradigm, not a new constraint or concentration result. The abstract now explicitly frames the contribution as methodological rather than statistical.
>
> W4: 90 hand-crafted prompts with predetermined difficulty
>
> Section 10.6 adds a benchmark evaluation on 150 prompts (50 each from TriviaQA, HotpotQA, TruthfulQA). Difficulty groups correspond to source datasets, with measured avg R(x) values (0.03, 0.40, 0.92) confirming the grouping aligns with intrinsic risk levels rather than being imposed by design. The verifier is LLaMA-3.3-70B in deterministic mode comparing generations against benchmark reference answers, not a rule-based heuristic. Table 7 reports three-way certification, Tables 8-9 compare baselines. CCI's certification rate is invariant across all four noise configurations; SC-SP violation rates on Hard inputs rise from 44% to 79% under false-negative noise, and Conf-SP approaches or exceeds ε=0.4 on Medium inputs (38-42%) across noise configurations.
>
> Structural comment: Theorem before Algorithm
>
> Algorithm 1 is now in Section 7 and theorems in Section 8, as requested.
>
> We are happy to provide further clarification if needed.

---

### Author Response · Authors · 2026-04-16
**Follow-up on revision**

We thank the reviewers and AE again for the valuable feedback and the opportunity to revise the manuscript. We have addressed all reviewer comments and incorporated the suggested clarifications and additions in the latest version.

We would like to kindly check if any further input or revisions are needed from our side at this stage. We are happy to provide additional clarifications or updates if helpful.

Thank you for your time and consideration.

---

> ### Author Response · Authors · 2026-05-19
> **Follow-up on review status**
>
> We hope the AE and reviewers are doing well.
>
> We would like to kindly follow up regarding the current status of our submission. We have addressed all reviewer comments in the revised manuscript and have not observed further updates since our previous follow-up.
>
> We would greatly appreciate any guidance on the next steps or whether additional input is needed from our side.
>
> Thank you for your time and consideration.